# Quantized Training of Gradient Boosting Decision Trees

**Yu Shi**[1]* **Guolin Ke**[2]*†,
**Zhuoming Chen**[3]‡ **Shuxin Zheng**[1] **Tie-Yan Liu**[1]
[1]Microsoft Research  [2]DP Technology  [3]Tsinghua University
yushi2@microsoft.com, kegl@dp.tech
chen-zm19@mails.tsinghua.edu.cn, {shuxin.zheng, tyliu}@microsoft.com

## Abstract

Recent years have witnessed significant success in Gradient Boosting Decision Trees (GBDT) for a wide range of machine learning applications. Generally, a consensus about GBDT's training algorithms is gradients and statistics are computed based on high-precision floating points. In this paper, we investigate an essentially important question which has been largely ignored by the previous literature - *how many bits are needed for representing gradients in training GBDT?* To solve this mystery, we propose to quantize all the high-precision gradients in a very simple yet effective way in the GBDT's training algorithm. Surprisingly, both our theoretical analysis and empirical studies show that the necessary precisions of gradients without hurting any performance can be quite low, e.g., 2 or 3 bits. With low-precision gradients, most arithmetic operations in GBDT training can be replaced by integer operations of 8, 16, or 32 bits. Promisingly, these findings may pave the way for much more efficient training of GBDT from several aspects: (1) speeding up the computation of gradient statistics in histograms; (2) compressing the communication cost of high-precision statistical information during distributed training; (3) the inspiration of utilization and development of hardware architectures which well support low-precision computation for GBDT training. Benchmarked on CPUs, GPUs, and distributed clusters, we observe up to $2\times$ speedup of our simple quantization strategy compared with SOTA GBDT systems on extensive datasets, demonstrating the effectiveness and potential of the low-precision training of GBDT. The code will be released to the official repository of LightGBM.[4]

## 1 Introduction

Gradient Boosting Decision Trees (GBDT) is a powerful machine learning algorithm. Despite the success of deep learning in recent years, GBDT is one of the best off-the-shelf choices of machine learning algorithms in many real-world tasks, including online advertising [33], search ranking [26, 14], finance prediction [29], etc. Along with carefully designed tools, including XGBoost [5], LightGBM [17], and CatBoost [24], GBDT has shown outstanding performance in various data science competitions [36] and industrial applications [32, 9, 28, 23].

Despite the success of GBDT, we found there is room for GBDT algorithms to fully exploit modern computation resources. First, the training of GBDT requires arithmetic operations of high-precision FP (Floating Point) numbers, which hinders the usage of low-precision computation resources. Low-precision training has become a standard technique to significantly accelerate the training of

---

*Corresponding authors.

†Work primarily done at Microsoft Research.

‡Work done during internship at Microsoft Research.

[4]https://github.com/Microsoft/LightGBM

neural networks [20]. The design of new processors for machine learning tends to encourage high throughput for low-precision computation. This results in a gap between GBDT training algorithms and the hardware architectures. Second, distributed training of GBDT is required for large datasets. The communication cost for distributed training, however, is huge due to high-precision statistical information passing among machines. The cost is especially high with a large number of features in the dataset, which hurts the scalability of GBDT in distributed systems.

In this paper, we propose a low-precision training algorithm for GBDT, based on stochastic gradient quantization. The main computation cost of training GBDT is the arithmetic operations over gradients. Before training each decision tree, we quantize the gradients into very low-precision (e.g., 2-bit or 3-bit) integers. Thus, we replace most of the FP arithmetic operations with integer arithmetic operations, which reduces computation time. In addition, low-bitwidth gradients result in a smaller memory footprint and better cache utilization. Techniques are proposed to preserve the accuracy of the model, including stochastic rounding in gradient quantization, and leaf-value refitting with original gradient values. We show both empirically and theoretically that quantized training of GBDT with low-bitwidth gradients is almost lossless in terms of model accuracy. Thus, we empower GBDT to utilize low-precision computation resources.

Distributed training of GBDT relies on the synchronization of gradient statistics. The gradient statistics are summarized into a histogram for each feature. With quantized gradients, only integer histogram values are needed in our algorithm. The communication among hosts or GPUs becomes sending and reducing of these integer values. And the size of these histograms is at least half of the original histograms with FP values. So quantized training of GBDT has an advantage in communication cost by nature and improves the scalability of GBDT on distributed systems.

We implement our system on both CPUs and GPUs. The results show that our methods accelerate the training of GBDT in various settings, including a single process on CPUs, a single process on a GPU, and distributed training over CPUs. This validates that our algorithm is general for different types of computation resources. Besides acceleration on existing hardware architectures, our algorithm also lay the foundation for GBDT to exploit new hardware architectures with more flexible support for low-precision operations in the future. Based on LightGBM, we implement a quantized GBDT training system. With quantized training, we achieve up to $2\times$ speedup compared with SOTA GBDT tools. Experiments also show that quantized GBDT training scales better in distributed systems.

Our algorithm also verifies huge redundancy in the information contained by gradient values in GBDT. Moreover, we reveal both theoretically and empirically how such redundancy can be properly reduced, to keep good model performance and training efficiency. And we show how to implement the quantized training of GBDT efficiently across different computation resources. We believe that these interesting findings will inspire new understandings and improvements for GBDT in the future.

## 2   Related Work

Trials to use low-precision training in deep neural networks (NN) are abundant. Using low bitwidth numbers in NN training can reduce memory access and accelerate training [13, 15, 35, 20, 10]. In addition to efficiency, the effectiveness of quantized NN training has also been discussed [7, 18, 4]. Besides quantization during the forward and backward calculations, quantization of gradients has also been applied in distributed training of neural networks [30] to reduce communication cost. However, the possibility of low-precision training for GBDT is rarely discussed in existing literature. RatBoost [22] is a boosting algorithm based on re-weighting samples. In RatBoost [22], quantization is applied to the weights of training samples during boosting, while the application of quantization in GBDT is not discussed. Recently gradient quantization is applied in federated training of GBDT [6] to transform FP gradients into large integers (e.g., 53-bit integers) for the sake of encryption. But quantization towards a very small number of bits is not explored. Accelerating GBDT training with quantized gradients was exploited by BitBoost [8]. And BitBoost shows empirically that it is possible to achieve comparable accuracy with quantized gradients of low bitwidth. However, BitBoost quantizes gradients in a deterministic way. In this paper, we provide a theoretical guarantee of quantized training based on stochastic quantization. And we show that stochastic quantization is crucial to preserve good model accuracy. Besides, BitBoost focuses on accelerating GBDT with bit operations on a single CPU core, and the implementation is a single-thread version. In this paper, we discuss important techniques in system implementation that enable significant acceleration across different computation platforms, including multi-core CPUs, GPUs, and distributed clusters.

Distributed training of GBDT is necessary for large-scale datasets. Training data are partitioned either by rows (data samples) or by columns (features) across different machines. We call the former strategy data-distributed training and the latter feature-distributed training. The communication cost for feature-distributed training grows linearly with the number of training samples [1, 19]. On the other hand, the communication cost for data-distributed training is proportional to the number of features [1, 19]. The two strategies can be combined with each machine getting only part of the features and samples [11]. The communication complexity of these strategies is analyzed in detail [31], with empirical studies [11]. Since training of GBDT requires summarizing the gradient statistics across the dataset into a histogram for each feature, the cost for data-distributed training is reduced with the more concise histogram information [5]. Thus, data-distributed training is more favorable with large-scale datasets. With many features, feature-distributed or the combination of two strategies can be faster [11].

Based on these distributed training strategies, some efforts have been made to reduce the communication cost. Meng et al [19] proposed Parallel-Voting Tree (PV-Tree) to further reduce the cost of data-distributed training. However, PV-Tree guarantees good performance only when different parts of the partitioned dataset have similar statistical distributions, which can require expensive random shuffling over the whole large dataset. Moreover, quantized training can be applied along with PV-Tree. This is because PV-Tree reduces communication cost by reducing the number of features to synchronize the statistics in the histograms. Since quantized training can reduce the size of histograms, it brings a general benefit for both PV-Tree and ordinary data-distributed training. DimBoost [16] keeps the number of communicated histograms unchanged but reduces the size of each histogram by compressing the histograms instead. Histograms represented by floating-point numbers in DimBoost are compressed into low-precision values (8-bit integers) before sending, and decoding into high-precision values. Note that in DimBoost, only the communication message for distributed training is in low-precision and the compression is lossy. In addition, reduction of the histograms still requires floating-point additions. In comparison, we quantized most parts of the training process of GBDT with low-precision integers, and only need to maintain histograms with integer values.

In this paper, we compare our algorithm with SOTA efficient implementations of GBDT, including XGBoost [5], LightGBM [17], and CatBoost [24]. They support training on CPUs, GPUs [34], and distributed training. However, none of these support low-precision training. We show that with quantized gradients, GBDT can be trained much faster compared with these SOTA tools in both single process and distributed settings, on both CPUs and GPUs. We implement quantized training based on LightGBM. But our method is general and can be adopted by all these tools.

## 3 Preliminaries of GBDT Algorithms

Gradient Boosting Decision Trees (GBDT) is an ensemble learning algorithm that trains decision trees as base components iteratively. In each iteration, for each training sample, the gradient (first-order derivative) and hessian (second-order derivative) of the loss function w.r.t. the current prediction value is calculated. Then a decision tree is trained to fit the negatives of gradients. Formally, in iteration $k + 1$, let $\hat{y}_i^k$ be the current prediction value of sample $i$. And $g_i$ and $h_i$ are the gradient and hessian of loss function $l$:

$$g_i = \frac{\partial l(\hat{y}_i^k, y_i)}{\partial \hat{y}_i^k}, h_i = \frac{\partial^2 l(\hat{y}_i^k, y_i)}{\left(\partial \hat{y}_i^k\right)^2} \tag{1}$$

For a leaf $s$, denote $I_s$ as the set of data indices in the leaf. Let $G_s = \sum_{i \in I_s} g_i$ and $H_s = \sum_{i \in I_s} h_i$ be the summations of $g_i$'s and $h_i$'s over samples in leaf $s$. With the tree structure being fixed in iteration $k + 1$, the training loss can be approximated by second-order Taylor polynomial:

$$\mathcal{L}_{k+1} \approx \mathcal{C} + \sum_s \left( \frac{1}{2} H_s w_s^2 + G_s w_s \right) \tag{2}$$

where $\mathcal{C}$ is a constant and $w_s$ is the prediction value of leaf $s$. By minimizing the approximated loss, we obtain the optimal value for leaf $s$ and the minimal loss contributed by data in $I_s$:

$$w_s^* = -\frac{G_s}{H_s}, \quad \mathcal{L}_s^* = -\frac{1}{2} \cdot \frac{G_s^2}{H_s}. \tag{3}$$

Finding the optimal tree structure is difficult. Thus the tree is trained by greedily and iteratively splitting a leaf into two child leaves, starting from a single root leaf. When splitting leaf $s$ into two

children $s_1$ and $s_2$ in tree $T_{k+1}$, we can calculate the reduction in the approximated loss

$$\Delta \mathcal{L}_{s \to s_1, s_2} = \mathcal{L}_s^* - \mathcal{L}_{s_1}^* - \mathcal{L}_{s_2}^* = \frac{G_{s_1}^2}{2H_{s_1}} + \frac{G_{s_2}^2}{2H_{s_2}} - \frac{G_s^2}{2H_s}. \tag{4}$$

To find the best split condition for leaf $s$, all split candidates of all features should be enumerated and the one with the largest loss reduction is chosen.

To accelerate the best split-finding process, an algorithm based on histograms is used by most state-of-the-art GBDT toolkits. The basic idea of histogram based GBDT is to divide the values of a feature into bins. Each bin corresponds to a range of feature values. Then we construct histograms with the bins such that each bin records the summation of gradients and hessians of data in that bin. Only the boundaries of the bins (ranges) will be considered as candidates of split thresholds. Algorithm 1 describes the process of histogram construction. Bin data matrix **data** records for each feature and sample the index of the bin in the histogram, i.e., which range the feature value falls in. The $g_i$'s and $h_i$'s are accumulated in the corresponding bins in histograms of the features. Since the split

---

**Algorithm 1** Histogram Construction for Leaf $s$

---

**Input**: Gradients $\{g_1, ..., g_N\}$, Hessians $\{h_1, ..., h_N\}$
**Input**: Bin data $\mathbf{data}[N][J]$, Data indices in leaf $s$ denoted by $I_s$
**Output**: Histogram $\mathbf{hist}_s$
**for** $i \in I_s, j \in \{1...J\}$ **do**
    $bin \leftarrow \mathbf{data}[i][j]$
    $\mathbf{hist}_s[j][bin].g \leftarrow \mathbf{hist}_s[j][bin].g + g_i$
    $\mathbf{hist}_s[j][bin].h \leftarrow \mathbf{hist}_s[j][bin].h + h_i$
**end for**

---

criterion (4) only depends on the summation of gradient statistics, we can easily obtain the optimal split threshold by iterating over bins in the histogram. Traditionally 32-bit floating point numbers are used for $g_i$'s and $h_i$'s. And the accumulations in the histograms usually require 32-bit or 64-bit FP numbers. In the next section, we show how to quantizes $g_i$'s and $h_i$'s into low-bitwidth integers so that most of the arithmetic operations can be replaced with integer operations with a fewer number of bits, which substantially saves computational cost.

## 4 Quantized Training of GBDT

We describe our quantized training algorithm for GBDT. First, we show the overall framework. Then we identify the two critical techniques to preserve the accuracy of quantized training, including stochastic gradient quantization and leaf-value refitting.

### 4.1 Framework for Quantized Training

We first quantize $g_i$ and $h_i$ into low-bitwidth integers $\widetilde{g}_i$ and $\widetilde{h}_i$. We divide the ranges of $g_i$'s and $h_i$'s of all training samples into intervals of equal length. To use $B$-bit ($B \geq 2$) integer gradients, we use $2^B - 2$ intervals. Each end of the intervals corresponds to an integer value, resulting in $2^B - 1$ integer values in total. Since the first-order derivatives $g_i$'s can take both positive and negative values, half of the intervals will be allocated for the negative values, and the other half for the positive values. For the second-order derivatives $h_i$'s, almost all commonly used loss functions of GBDT have non-negative values. We assume that $h_i \geq 0$ in subsequent discussions. Thus the interval lengths are

$$\delta_g = \frac{\max_{i \in [N]} |g_i|}{2^{B-1} - 1}, \quad \delta_h = \frac{\max_{i \in [N]} h_i}{2^B - 2} \tag{5}$$

for $g_i$'s and $h_i$'s, respectively. Then we can calculate the low-bitwidth gradients

$$\widetilde{g}_i = Round\left(\frac{g_i}{\delta_g}\right), \quad \widetilde{h}_i = Round\left(\frac{h_i}{\delta_h}\right). \tag{6}$$

The function $Round$ rounds a floating point number into an integer number. Note that in the case where $h_i$'s are constant, there's no need to quantize $h_i$. We left the detailed rounding strategy in Section 4.2. We replace $g_i$'s and $h_i$'s in Algorithm 1 with $\widetilde{g}_i$'s and $\widetilde{h}_i$'s. Then addition operations of the original gradients can be directly replaced by integer additions. Concretely, the statistics $g$ and $h$ in the histogram bins will become integers. Thus accumulating gradients into histogram bins in

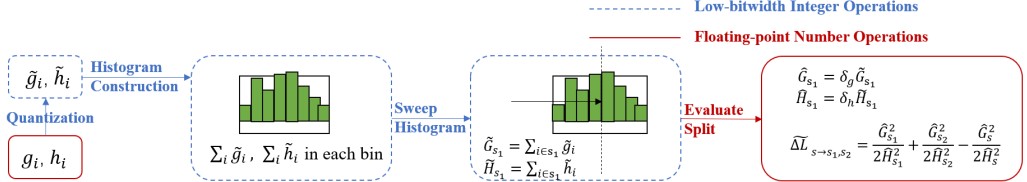

Figure 1: Workflow of the quantized GBDT training.

Algorithm 1 only requires integer additions. In equation (4), $G_{s_1}$, $H_{s_1}$, $G_{s_2}$ and $H_{s_2}$ will replaced by their integer counterparts $\widetilde{G}_{s_1}$, $\widetilde{H}_{s_1}$, $\widetilde{G}_{s_2}$, and $\widetilde{H}_{s_2}$. We found that 2 to 4 bits for quantized gradients are enough for good accuracy. As we will discuss in Section 6.1 and 7.4.3, in most cases, 16-bit integers will be enough to accumulate such low-bitwidth gradients in histograms. Thus, most of the operations are done with low-bitwidth integers. We only need floating point operations when calculating the original gradients, hessians and the split gain. Specifically, the split gain is estimated as

$$\Delta\widetilde{\mathcal{L}}_{s \to s_1, s_2} = \frac{\left(\widetilde{G}_{s_1}\delta_g\right)^2}{2\widetilde{H}_{s_1}\delta_h} + \frac{\left(\widetilde{G}_{s_2}\delta_g\right)^2}{2\widetilde{H}_{s_2}\delta_h} - \frac{\left(\widetilde{G}_s\delta_g\right)^2}{2\widetilde{H}_s\delta_h}, \tag{7}$$

where the scale of the gradient statistics is recovered by multiplying $\delta_g$ and $\delta_h$. Figure 1 summarizes the workflow for quantized GBDT training.

## 4.2 Rounding Strategies and Leaf-Value Refitting

We find that the $Round$ strategy in equation (6) has a significant impact on the accuracy of the quantized training algorithm. Rounding to the nearest integer number seems to be a reasonable choice. However, we found the accuracy drops severely with this strategy, especially with a small number of gradient bits. Instead, we adopt a stochastic rounding strategy. $\widetilde{g}_i$ randomly takes values $\lfloor g_i/\delta_g \rfloor$ or $\lceil g_i/\delta_g \rceil$ such that $\mathbb{E}[\widetilde{g}_i] = g_i/\delta_g$. The formal definitions of the two rounding strategies are

$$\text{RN}(x) = \begin{cases} \lfloor x \rfloor, & x < \lfloor x \rfloor + \frac{1}{2} \\ \lceil x \rceil, & x \geq \lfloor x \rfloor + \frac{1}{2} \end{cases}, \quad \text{SR}(x) = \begin{cases} \lfloor x \rfloor, & \text{w.p.} \quad \lceil x \rceil - x \\ \lceil x \rceil, & \text{w.p.} \quad x - \lfloor x \rfloor \end{cases} \tag{8}$$

where we use RN for round-to-nearest and SR for stochastic rounding. The key insight is that the split gain is calculated with the summation of gradients. And stochastic rounding would provide an unbiased estimation for the summations, i.e. $\mathbb{E}[\widetilde{G}\delta_g] = G$, and $\mathbb{E}[\widetilde{H}\delta_h] = H$ in equation (7). The importance of stochastic rounding is also recognized in other scenarios for quantization, including NN quantization training [13] and histogram compression in DimBoost [16]. With stochastic rounding, we provide a theorem in Section 5 which ensures that the error in split gain estimation caused by quantized gradients is bounded by a small value with high probability.

With quantized gradients, the optimal leaf value in equation (3) becomes $\widetilde{w}_s^* = -\frac{\widetilde{G}_s\delta_g}{\widetilde{H}_s\delta_h}$. In most cases, $\widetilde{w}_s^*$ is enough for good results. But we found that for some loss functions, especially ranking objectives, refitting the leaf values with the original gradients after the tree has stopped growing is useful to improve the accuracy of quantized training. BitBoost [8] also recalculates leaf values with real gradients after tree growing. The difference is that here we consider hessians in split gain (7) during tree growth, but BitBoost treats hessians as constants during tree growth and uses true hessians only when refitting the leaf values. It is cheap to calculate $w_s^*$ in equation (3) given a fixed tree structure since only a single pass over $g_i$'s and $h_i$'s is needed to sum them up into different leaves.

## 5 Theoretical Analysis

We prove that the difference in the gain of a split due to gradient quantization is bounded by a small value, with enough training data. We consider loss functions with constant second-order derivatives (i.e., the $h_i$'s are constant). The square error loss function $l(\hat{y}, y) = \frac{1}{2}(\hat{y} - y)^2$ for regression tasks falls in this category. The analysis is also fit for GBDT without second-order Taylor approximation. We leave the case for loss functions with non-constant second-order derivatives in Appendix B. We focus on the discussion of the conclusion in this section and leave details of the proof in Appendix B.

Our theoretical analysis is based on a more specific form of weak-learnability assumption in boosting. Specifically, the assumption considers only two-leaf decision trees (a.k.a. stumps) as weak learners.

**Definition 5.1** (Weak Learnability of Stumps) Given a binary classification dataset $\mathcal{D} = \{(\mathbf{x}_i, c(\mathbf{x}_i))\}_{i=1}^N$ where $c(\mathbf{x}_i) \in \{-1, 1\}$, weights $\{w_i\}_{i=1}^N$, $w_i \geq 0$ and $\sum_i w_i > 0$, there exists $\gamma > 0$ and a two-leaf decision tree with leaf values in $\{-1, 1\}$ s.t. the weighted classification error rate on $\mathcal{D}$ is $\frac{1}{2} - \gamma$. Then the dataset $\mathcal{D}$ is $\gamma$-empirically weakly learnable by stumps w.r.t. $c$ and $\{w_i\}_{i=1}^N$.

The weak learnability of stump states that given a binary-class concept $c$ and weights for dataset $\mathcal{D}$, there exists a stump that can produce slightly better classification accuracy than a random guess. Our analysis is based on the following assumption over the data in a single leaf $s$.

**Assumption 5.2** Let $\text{sign}(\cdot)$ be the sign function (with $\text{sign}(0) = 1$). For data subset $\mathcal{D}_s \subset \mathcal{D}$ in leaf $s$, there exists a stump and a $\gamma_s > 0$ s.t. $\mathcal{D}_s$ is $\gamma_s$-empirically weakly learnable by stumps, w.r.t. concept $c(\mathbf{x}_i) = \text{sign}(g_i)$ and weights $w_i = |g_i|$, where $i \in I_s$.

An equivalence of the assumption is: $\mathcal{D}$ is $\gamma_s$-empirically weakly learnable by stumps, w.r.t. concept $c(\mathbf{x}_i) = \text{sign}(g_i)$, and weights $w_i = |g_i|$ for $i \in I_s$ and $w_i = 0$ for $i \in [N] \backslash I_s$. A similar assumption has been adopted in previous theoretical analysis for gradient boosting [12]. The difference is that here we restrict the weak learner to be a stump. In Appendix B, we show that for any leaf $s$ with positive split gain, there exists such a $\gamma_s > 0$. As we will see in the experiments, for most leaves during the training of GBDT, $\gamma_s$ won't be too small.

Let $\mathcal{G}_{s \rightarrow s_1, s_2} = -\left(\mathcal{L}_{s_1}^* + \mathcal{L}_{s_2}^*\right) \geq 0$, where $\mathcal{L}_{s_1}^*$ and $\mathcal{L}_{s_2}^*$ are defined in equation (3). And let $\widetilde{\mathcal{G}}_{s \rightarrow s_1, s_2} = -\left(\widetilde{\mathcal{L}}_{s_1}^* + \widetilde{\mathcal{L}}_{s_2}^*\right) \geq 0$ be the estimated version of $\mathcal{G}_{s \rightarrow s_1, s_2}$ after gradient quantization. And let $\mathcal{G}_s^* = \mathcal{G}_{s \rightarrow s_1^*, s_2^*}$ be the value for the optimal split $s \rightarrow s_1^*, s_2^*$ in leaf $s$. Note that for leaf $s$, the optimal split is chosen only according to $\mathcal{G}_{s \rightarrow s_1, s_2}$, since $\mathcal{L}_s^*$ is a constant for different splits of leaf $s$. Then based on Assumption 5.2, we have the following theorem.

**Theorem 5.3** For loss functions with constant hessian value $h > 0$, if Assumption 5.2 holds for the subset $\mathcal{D}_s$ in leaf $s$ for some $\gamma_s > 0$, then with stochastic rounding and leaf-value refitting, for any $\epsilon > 0$, and $\delta > 0$, at least one of the following conclusions holds:

1. With any split of leaf $s$ and its descendants, the resultant average of absolute values of prediction values by the tree in the current boosting iteration for data in $\mathcal{D}_s$ is no greater than $\epsilon/h$.

2. For any split $s \rightarrow s_1, s_2$ of leaf $s$, with a probability of at least $1 - \delta$,

$$\frac{\left|\widetilde{\mathcal{G}}_{s \rightarrow s_1, s_2} - \mathcal{G}_{s \rightarrow s_1, s_2}\right|}{\mathcal{G}_s^*} \leq \frac{\max\limits_{i \in [N]} |g_i| \sqrt{2 \ln \frac{4}{\delta}}}{\gamma_s^2 \epsilon \cdot 2^{B-1}} \left(\sqrt{\frac{1}{n_{s_1}}} + \sqrt{\frac{1}{n_{s_2}}}\right) + \frac{\left(\max\limits_{i \in [N]} |g_i|\right)^2 \ln \frac{4}{\delta}}{\gamma_s^2 \epsilon^2 n_s \cdot 4^{B-2}}. \quad (9)$$

In equation (9), the bound of gradients $\max_{i \in [N]} |g_i|$ is determined by the labels and the loss function. For regression tasks, we can normalize the labels to a fixed range, e.g., $[0, 1]$, thus the gradients are bounded. For classification tasks, the gradients are bounded (in the range $[-1, 1]$) by nature with cross-entropy loss. With enough data in leaf $s$, if the split is not too imbalanced and partitions enough data into both children $s_1$ and $s_2$, then we can expect the error caused by quantization is small. Theorem 5.3 ensures that with quantized training, either a split of a leaf has a limited impact on the prediction values, or quantization does not change the gains of the splits too much.

In equation (9), we also notice that with more bits (larger $B$) to represent gradients in quantization, the error will be smaller. This is consistent with our intuition that with more quantized values, we can approximate the original gradients better. But in practice, we found that a smaller number of bits (e.g., 2 to 4 bits) is enough for good accuracy. With $n_s = 10^7$ in parent node $s$, and $n_{s_1} = n_{s_2} = 5 \times 10^6$. Let $\epsilon = \max_{i \in [N]} |g_i| / 10$, assume that in leaf $s$ we have $\gamma_s = 0.2$ and let $\delta = 0.1$. With 3 bits for gradients, the upper bound in (9) is approximately 0.15. With 4 bits, the value is smaller than 0.08. Thus, with a high probability (greater than 90%), changes in split gain due to quantization is well bounded.

We have a similar analysis for the cases where hessians $h_i$'s are not constant. For non-constant hessian values, the analysis requires additional assumptions. Due to limited space, we left the details in Appendix B.

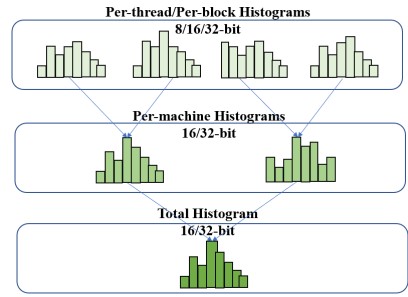

Figure 2: Hierarchical histogram buffers.

Table 1: Datasets used in experiments.

| Name | #Train | #Test | #Attribute | Task | Metric |
|---|---|---|---|---|---|
| Higgs [2] | 10,500,000 | 500,000 | 28 | Binary | AUC↑ |
| Epsilon[5] | 400,000 | 100,000 | 2,000 | Binary | AUC ↑ |
| Kitsune [21] | 14,545,195 | 3,999,993 | 115 | Binary | AUC↑ |
| Criteo[6] | 41,256,555 | 4,584,061 | 67 | Binary | AUC↑ |
| Bosch[7] | 1,000,000 | 183,747 | 968 | Binary | AUC↑ |
| Year [3] | 463,715 | 51,630 | 90 | Regression | RMSE↓ |
| Yahoo LTR [8] | 473,134 | 165,660 | 700 | Ranking | NDCG@10↑ |
| LETOR [25] | 2,270,296 | 753,611 | 137 | Ranking | NDCG@10↑ |

## 6 System Implementation

In this section, we show how to implement quantized GBDT for good efficiency. Current CPU and GPU architectures have limited support for the representation and calculation of low-precision numbers. For example, most CPUs by today only support 8-bit integers as the smallest data type. Under these limitations, the benefits of quantized training cannot be fully exploited. Though the gradients can be compressed into integers with only 2 to 4 bits, we must use at least 8-bit arithmetics for accumulations in histograms. Even with these limitations, we achieve a considerable speedup on existing CPU and GPU architectures, given the techniques introduced in this section.

### 6.1 Hierarchical Histogram Buffers

Note that in Algorithm 1 most operations are accumulating gradients. Thus, low-bitwidth gradients will not bring much speedup if high-bitwidth integers are used to store the accumulations. On the contrary, integer overflow may occur with low-bitwidth integers storing the accumulations. To maximally exploit low-precision computation resources while avoiding integer overflow, we partition the training dataset by rows (samples) and assign one partition to a thread on CPU or multiple CUDA blocks on GPU. Thus, each thread or CUDA block constructs a local histogram over the samples in the assigned partition. Since the number of training samples per partition is much smaller, in most cases 16-bit integers for the accumulations in the histogram is enough. In the end, local histograms are reduced to the total histogram which may use more bits per bin, e.g. with 16-bit or 32-bit integers. Figure 2 displays this divide-and-merge strategy and how the bitwidths differ in local and total histograms.

### 6.2 Packed Gradient and Hessian

We pack the summations of gradients and hessians in a histogram bin into a single integer in a similar way as SecureBoost+ [6]. For example, if we use two 16-bit integers to store the accumulation values, then the packed accumulation value would be a single 32-bit integer. In our case, however, since gradients can be either negative or positive, the accumulation of gradients must reside in the upper part of the packed integer to exploit the signed bit. The accumulation of hessians resides in the lower part of the packed integer. When accumulating the gradient and hessian of a sample into a bin, the low-bitwidth gradient and hessian are packed in the same way before the addition.

Packed gradient and hessian halves the number of memory accesses in histogram construction. It also halves the number of integer additions. On GPUs, the packing has one more contribution to efficiency. This is because histogram construction on GPUs relies on atomic operations to ensure correctness when multiple threads add to the same histogram. Thus, packing also reduces the overhead of atomic operations.

## 7 Experiments

We evaluate our quantized GBDT training system on public datasets and compare both accuracy and efficiency with other GBDT tools. First, we show that quantized GBDT training preserves accuracy with low-bitwidth gradients. Then, we assess the efficiency on CPUs and GPU of a single machine,

---

[5] https://www.csie.ntu.edu.tw/ cjlin/libsvmtools/datasets/binary.html#epsilon

[6] https://go.criteo.net/criteo-research-kaggle-display-advertising-challenge-dataset.tar.gz

[7] https://www.kaggle.com/c/bosch-production-line-performance

[8] https://webscope.sandbox.yahoo.com/catalog.php?datatype=c

Table 2: Comparison of accuracy, w.r.t. different quantized bits.

| Algorithm | Binary Classification | | | | | Regression | Ranking | |
|---|---|---|---|---|---|---|---|---|
| | Higgs↑ | Epsilon↑ | Kitsune↑ | Criteo↑ | Bosch↑ | Year↓ | Yahoo LTR↑ | LETOR↑ |
| XGBoost | 0.845778 | 0.950210 | 0.948329 | 0.802030 | **0.706423** | 8.954460 | **0.794919** | 0.505058 |
| CatBoost | 0.845425 | 0.943211 | 0.944557 | 0.803150 | 0.687795 | 8.951745 | 0.794215 | 0.519952 |
| LightGBM | 0.845694 | 0.950203 | 0.950561 | 0.803791 | 0.703471 | 8.956278 | 0.793792 | 0.524191 |
| 2-bit $SR_{refit}$ | 0.845587 | 0.949472 | 0.952703 | 0.803293 | 0.700040 | 8.953388 | 0.788579 | 0.519067 |
| 3-bit $SR_{refit}$ | 0.845725 | 0.949884 | 0.951309 | 0.803768 | 0.702025 | **8.937374** | 0.791077 | 0.522220 |
| 4-bit $SR_{refit}$ | 0.845507 | 0.950049 | 0.950911 | 0.803783 | 0.702959 | 8.942898 | 0.792664 | 0.523702 |
| 5-bit $SR_{refit}$ | 0.845706 | 0.950298 | 0.949229 | 0.803766 | 0.703242 | 8.948542 | 0.793166 | **0.524616** |
| 2-bit $SR_{no\ refit}$ | **0.846713** | 0.944509 | 0.952974 | 0.803750 | 0.701399 | 9.112302 | 0.764862 | 0.486193 |
| 3-bit $SR_{no\ refit}$ | 0.846040 | 0.949593 | 0.951385 | **0.803922** | 0.702460 | 8.990034 | 0.780041 | 0.507689 |
| 4-bit $SR_{no\ refit}$ | 0.845816 | 0.950127 | 0.951197 | 0.803812 | 0.704053 | 8.955256 | 0.787575 | 0.515767 |
| 5-bit $SR_{no\ refit}$ | 0.845842 | 0.950275 | 0.949794 | 0.803790 | 0.702717 | 8.952768 | 0.791631 | 0.520900 |
| 2-bit $RN_{refit}$ | 0.795991 | 0.889149 | 0.962201 | 0.779906 | 0.685407 | 9.429014 | 0.765103 | 0.454512 |
| 3-bit $RN_{refit}$ | 0.830506 | 0.944329 | **0.966606** | 0.782732 | 0.688372 | 9.062854 | 0.772364 | 0.476874 |
| 4-bit $RN_{refit}$ | 0.840747 | 0.949946 | 0.961938 | 0.795803 | 0.691163 | 8.968694 | 0.777347 | 0.487394 |
| 5-bit $RN_{refit}$ | 0.843820 | **0.950457** | 0.962427 | 0.802438 | 0.698529 | 8.952418 | 0.784333 | 0.494828 |
| 2-bit $RN_{no\ refit}$ | 0.836683 | 0.925220 | 0.946016 | 0.768338 | 0.695089 | 10.685840 | 0.632058 | 0.203732 |
| 3-bit $RN_{no\ refit}$ | 0.843482 | 0.946850 | 0.940961 | 0.791709 | 0.697933 | 9.377560 | 0.732487 | 0.350127 |
| 4-bit $RN_{no\ refit}$ | 0.845788 | 0.949676 | 0.949228 | 0.802689 | 0.702767 | 8.969828 | 0.765432 | 0.437317 |
| 5-bit $RN_{no\ refit}$ | 0.845765 | 0.950307 | 0.952420 | 0.803645 | 0.695559 | 8.965400 | 0.782608 | 0.485752 |

and scalability with distributed training over multiple CPU machines. Ablation study is provided to analyze the techniques for accuracy preservation and system implementation. Table 1 lists the datasets used for the experiments. For Criteo, we encode the categorical features by target and count encoding. We include open-source GBDT tools XGBoost, LightGBM, and CatBoost as baselines. We use the leaf-wise tree-growing strategy [27] for all baselines. A full description of datasets and hyperparameter settings is provided in Appendix C.

## 7.1 Accuracy of Quantized Training

We evaluate the accuracy of quantized training. For each dataset and each algorithm, the result of the best iteration of the test set is reported. We use gradients with 2-bit, 3-bit, 4-bit and 5-bit integers, and compare the results with 32-bit single-precision FP gradients. Table 2 lists the best metric achieved on the test set across all boosting iterations. We denote stochastic rounding by SR and round-to-nearest by RN. And use the subscripts to indicate whether to apply leaf-value refitting. Here we focus on the effect of quantization on accuracy only (the $SR_{refit}$ part), and leave the discussion of rounding strategy and leaf-value refitting in 7.4.1. And due to limited space, we list only metrics of the CPU version here. Appendix C provides a similar table for GPU version. With low-bitwidth gradients, the accuracy is not affected much. For some datasets, including Epsilon, Yahoo LTR, and LETOR, more gradient bits can improve the performance. Other datasets are less sensitive to the number of bits.

## 7.2 Speedup on Standalone Machine

We evaluate the acceleration brought by quantized training ($SR_{refit}$) on a single machine by comparing it with CPU and CUDA implementations of popular open-source GBDT tools. Our quantized training on CPU is based on LightGBM. We compare our CPU implementation with XGBoost, CatBoost, and LightGBM with 32-bit gradients. For the comparison on GPU, we implement a new CUDA version of LightGBM (denoted by LightGBM+, details are in Appendix E) and further implement quantized training based on it. We summarize the benchmark results in Table 3. Overall, quantized training achieves comparable accuracy with a shorter training time. For the histogram construction time, quantized training speeds up histogram construction by up to 3.8 times on GPU, with an overall speedup of up to 2.2 times compared with LightGBM+. We also see the acceleration on the CPU, but with relatively small gains compared with GPU. In summary, quantized training pushes forward the SOTA efficiency of GBDT algorithms. The number of bits for gradients does not influence the training time much. For histogram construction time, we only list the results with 2-bit gradients due to limited space. A full table can be found in Table 6 of Appendix C.

## 7.3 Speedup of Distributed Training

We evaluate quantization in distributed GBDT training on an enlarged version of the Epsilon dataset, which duplicates 20 copies of the training set of the original Epsilon dataset (denoted by Epsilon-8M),

Table 3: Detailed time costs for different algorithms in different datasets (seconds).

| | Algorithm | Higgs | Epsilon | Kitsune | Criteo | Bosch | Year | Yahoo LTR | LETOR |
|---|---|---|---|---|---|---|---|---|---|
| | XGBoost | 33.97 | 311.12 | 181.24 | 326.82 | 68.44 | 20.47 | 28.64 | 51.29 |
| | CatBoost | 61.10 | 105.00 | 80.20 | 187.80 | 22.12 | 33.96 | 59.22 | N/A |
| | LightGBM+ | 29.05 | 87.12 | 77.43 | 102.33 | 21.41 | 24.33 | 30.79 | 41.79 |
| GPU total time | LightGBM+ 2-bit | 24.78 | **39.04** | **38.26** | 61.04 | 12.57 | **18.19** | **23.09** | **33.60** |
| | LightGBM+ 3-bit | **24.45** | 39.25 | 38.63 | 59.93 | 12.60 | 18.24 | 24.93 | 33.87 |
| | LightGBM+ 4-bit | 24.53 | 39.82 | 40.00 | **59.49** | 12.55 | 18.34 | 25.65 | 34.11 |
| | LightGBM+ 5-bit | 24.55 | 41.30 | 40.83 | 60.24 | **12.08** | 18.41 | 25.50 | 34.36 |
| | XGBoost | 109.16 | 1282.97 | 281.72 | 565.52 | 130.92 | 28.85 | 103.87 | 72.37 |
| | CatBoost | 1009.8 | 1283.4 | 1495.0 | 7702.2 | 998.4 | 95.8 | 588.2 | 865.4 |
| | LightGBM | 83.27 | 519.89 | 332.12 | 524.61 | 59.94 | 12.67 | 75.44 | 103.09 |
| CPU total time | LightGBM 2-bit | 73.36 | **426.50** | 215.91 | 444.28 | 46.63 | 12.94 | 61.50 | **72.08** |
| | LightGBM 3-bit | 69.64 | 459.39 | **207.96** | 440.68 | 47.35 | 12.79 | **61.07** | 74.35 |
| | LightGBM 4-bit | **69.30** | 458.62 | 208.99 | **416.60** | **46.45** | 11.90 | 61.15 | 77.66 |
| | LightGBM 5-bit | 69.86 | 457.68 | 211.53 | 423.80 | 47.52 | **11.79** | 61.76 | 77.92 |
| GPU Hist. time | LightGBM+ | 11.26 | 46.96 | 54.77 | 70.97 | 16.57 | 9.61 | 11.59 | 17.75 |
| | LightGBM+ 2-bit | 4.84 | 12.11 | 16.41 | 21.74 | 8.52 | 4.08 | 8.23 | 10.20 |
| CPU Hist. time | LightGBM | 50.74 | 458.46 | 253.07 | 385.98 | 53.08 | 6.68 | 58.53 | 66.39 |
| | LightGBM 2-bit | 32.82 | 375.70 | 147.10 | 269.00 | 39.80 | 5.99 | 43.59 | 38.23 |

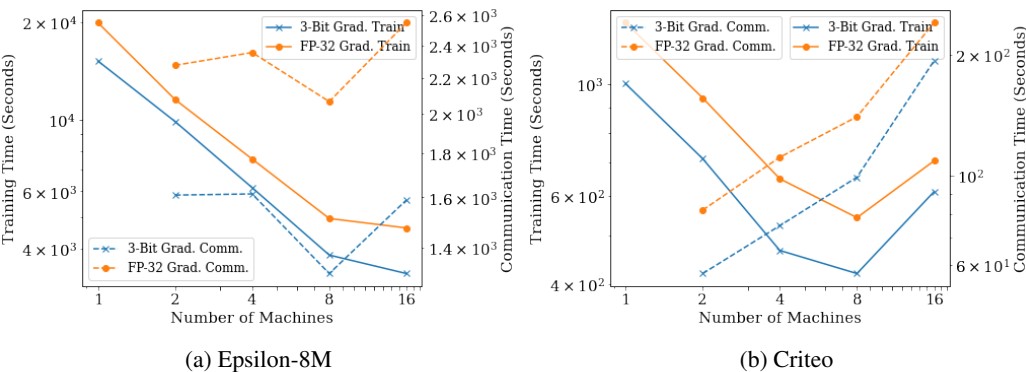

(a) Epsilon-8M                    (b) Criteo

Figure 3: Scaling on distributed systems.

and Criteo. Figure 3 shows how training time and communication time vary with different numbers of machines. Quantization can consistently reduce communication cost and accelerate distributed training. On the Criteo dataset, the communication cost dominates with 16 machines, and the training time with 16 machines is even slower than with 4 machines. This shows that there is still some room for improvement in the scalability of our system.

### 7.4 Ablation Study

We analyze the effect of proposed techniques that ensure the effectiveness and efficiency of quantized training. We also provide a discussion on the feasibility of the weak-learnability assumption used in Section 5.

#### 7.4.1 Rounding Strategies and Leaf-Value Refitting

Stochastic rounding plays an important role in the accuracy of quantized training. Table 2 also compares the accuracy on test sets with round-to-nearest (denoted by RN) and stochastic rounding (denoted by SR), with different numbers of gradient bits. For both strategies, the results with and without leaf-value refitting are both reported. With smaller numbers of bits, the SR strategy significantly outperforms the RN strategy. Note that leaf-value refitting is not able to compensate for the drawback of the RN. Table 2 also demonstrates how leaf-value refitting influences accuracy. For binary classification, leaf-value refitting is less important as a comparable accuracy could be achieved without it. While for regression and ranking tasks, leaf-value refitting is crucial to achieving comparable accuracy.

### 7.4.2 Packed Gradient and Hessian

Figure 5 shows the training time and histogram construction time with and without packing gradient and hessian, testing with 2-bit gradients of CPU version. No-packing version on GPU requires atomic addition for 16-bit integers, which is not natively supported by NVIDIA V100 GPUs. Thus, for a clear comparison, we only show the results of CPU version here. Overall, packing is important for the speed of histogram construction.

### 7.4.3 Histogram Bitwidth



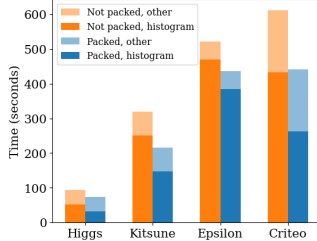

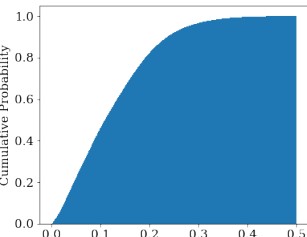

Figure 4: Histogram bitwidth frequency.   Figure 5: Speedup by packing gradients.   Figure 6: Cumulative distribution of $\hat{\gamma}_s$.

Figure 4 shows the frequency of per-thread histograms of different bitwidths used during training with 2-bit gradients on CPUs. The counts are the number of leaves in the boosting process using the corresponding bitwidth in per-thread histograms. On CPUs, our per-thread histograms can use either 8-bit, 16-bit, or 32-bit integers. For now 8-bit integer histogram is only available with 2-bit gradients in our current version. The bitwidth in histograms is determined by the number of training samples in the leaves, the number of threads/blocks to use, the total number of bins in histograms of all features, and the gradient bitwidth. The minimum bitwidth of histogram integers that won't result in an integer overflow is chosen. The figure shows that quantized training exploits low-precision computations. On GPUs, there's no native support for atomic additions of packed 8-bit integers. Thus, we use 16-bit histograms in CUDA blocks in our current implementation. We leave more aggressive usage of low-bitwidth computations in our future work.

### 7.4.4 Feasibility of Weak-Learnability Assumption

To verify the feasibility of Assumption 5.2, we should evaluate the largest $\gamma_s$ with which a stump with weighted classification error rate $\frac{1}{2} - \gamma_s$ exists. This requires enumerating all the splits again and calculating the weighted error rate for each split, which is costing. Luckily, we have an easy approach to obtaining a lower bound of the largest $\gamma_s$, which can be naturally recorded during training of GBDT, we name the lower bound as $\hat{\gamma}_s$. $\hat{\gamma}_s$ is the weighted error rate of the optimal split in leaf $s$ according to the criterion in (4). Figure 6 shows the cumulative distribution of $\hat{\gamma}_s$ of all leaves during quantized training of the Year dataset with 3-bit gradients. For most leaves (over 75%), a $\gamma_s > 0.05$ exists. Thus $\gamma_s$ won't be too small for most leaves.

## 8    Discussion and Future Work

In this paper, we try to answer an important but previously neglected question: Can GBDT exploit low-precision training? We propose a quantized training framework with low-bitwidth integer arithmetics which enables the low-precision training of GBDT. We identify the key techniques to preserve accuracy with quantized training, including stochastic rounding and leaf-value refitting. Theoretical analysis shows that quantization has a limited impact on the selection of optimal splits in leaves, given enough training data. We implement our algorithm with both CPU and GPU versions. Experiments show that our quantized training GBDT method can achieve comparable accuracy with significant speedup over SOTA GBDT tools, with CPUs, GPUs, and distributed clusters. In this work, we propose a simple stochastic quantization framework. Designing more sophisticated quantization methods for GBDT is an interesting direction for future exploration. We also leave the implementation of quantized GBDT training on multi-GPU distributed systems and more aggressive usage of low-bitwidth computations in our future work. We believe our method will bring new inspirations in improving GBDT training algorithms, as low-precision computation becomes a trend in machine learning.

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
