# OpenReview forum: "Quantized Training of Gradient Boosting Decision Trees"
_NeurIPS.cc/2022/Conference — NeurIPS 2022 Accept_

### Official Review · Reviewer_MfJi · 2022-07-06

**Rating:** 7
**Confidence:** 3
**Soundness:** 4 excellent
**Presentation:** 3 good
**Contribution:** 3 good

**Summary:**

The paper proposes a quantised version of Gradient Boosted Decision Trees. The paper proposes to quantise the high-precision gradients in a rather simple way. The paper considers the problem how many bits are really needed for gradients in GBDT to achieve a reasonable performance. It is shown that with the low-precision gradients most arithmetic operations may be replaced by integer operations.

**Questions:**

The last paragraph on page 3 is not extremely clearly written, e.g., what is meant under "And construct histograms with the bins"?

Maybe I missed it : the results reported in Tables 2 and 3 comparing various boosting algorithms: are the number of estimators and the depth of trees the same in all these methods? How many estimators (trees) and what is the depth?

**Limitations:**

It would be interesting to discuss the cases (if they exist) where quantised gradients lead to much worse performance than continuous ones.

**Strengths And Weaknesses:**

Strengths. The paper is clearly written. The results both theoretical and numerical are convincing.

Weaknesses. The idea to perform training with the quantised gradients is not new. However, the papers introducing the quantised learning which appeared 5-6 years ago (Binarised neural networks, results of M. Courbariaux and Y. Bengio) are well cited.

---

> ### Author Response · Authors · 2022-08-02
> **Response to Reviewer MfJi - Part 1**
>
> We sincerely thank the reviewer for reviewing our paper, supporting our work, and providing valuable suggestions. We address your concerns as follows:
>
> **Regarding the last paragraph on page 3** Thanks for the reminder. We will elaborate this paragraph to make it more clear. The paragraph describes the histogram-based training for GBDT. "construct histograms with the bins" means accumulating the gradients and hessians of data points into the buckets of feature histograms. For example, we divide the range of values of feature $j$ into 3 buckets $[0, 0.3]$, $[0.3, 0.7]$ and $[0.7, 1.0]$. If feature $j$ of data $x_i$ is $0.4$, then $g_i$ and $h_i$ will be accumulated in the second bin of the histogram of feature $j$, since $0.4$ is in the second bucket. With discretized version of feature values, the process can be described in Algorithm 1 in our paper.
>
> **Regarding settings of Table 2 and 3**
> Yes. The number of estimators and the maximum number of leaves per tree are the same for all methods for a fair comparison. We use 500 decision tree estimators for all methods. And we set the maximum number of leaves per tree to 255 for all methods. More details of our experiment settings can be found in Section C of our Appendix.

---

> ### Author Response · Authors · 2022-08-02
> **Response to Reviewer MfJi - Part 2**
>
> We sincerely thank the reviewer for reviewing our paper, supporting our work, and providing valuable suggestions. We address your concerns as follows:
>
> **Regrading the novelty.** Training models with quantized gradients for neural networks are well-studied in previous literature but little is known on tree models. The differences appear in motivation, challenges, theoretical analysis, and implementations.
> - **Motivation.**
> (1). Training NN with quantized gradients aims to reduce the communication cost in distributed deep learning to speed up the training[1][2][3]. While both computation and communication cost could be largely reduced in training GBDT with quantized gradients. This is because that the computational bottleneck of NN are forward and backward propagation, which involves both weights and gradients, but the calculation of gradients in GBDT is quite fast, and the computation bottleneck is calculation of the statistics of gradients in histograms, which are also the main content of communication in distributed training. Therefore, gradient quantization would both accelerate training and communication in GBDT.
> (2). Quantized gradient slows the convergence in NN training because it enlarges the gradient's variance and will affect the convergence speed stochastic optimization algorithms. While it has little effect on training of GBDT since the model only care about the statistics of gradients, and the summation of stochastic quantization gradients are unbiased approximation to the summation of full-precision gradients. Therefore, training with gradient quantization is more well-motivated for GBDT models. We will fully discuss the differences in the next version of our paper.
>
> - **Challenges.** Training GBDT with quantized gradients faces additional challenges. For example, sometimes the stochastic quantization could substantially changes the splitting of the leaf, and brings intractable accuracy drop. To overcome such challenge, we identify stochastic quantization and leaf-value refitting are two important techniques. And the techniques are carefully verified both theoretically and empirically.
>
> - **Theoretical Analysis.** To best of our knowledge, we're the first to provide comprehensive theoretical analysis for training GBDT with quantized gradients. The analysis is novel and non-trivial. It is intuitive that with more data, the estimated summation with stochastic quantized gradient will be more accurate. However, to prove that the estimated split gain with quantized gradients is an asymptotically consistent estimator of the original split gain is difficult. We find that the key to achieve this is to consider the weak learnability assumption (Assumption 5.1).
>
> - **Implementations.** We provide a complete training framework for quantized GBDT with high efficiency in practical both on CPU and GPU architectures. It's quite challenging because that the support of low-precision calculation is limited for both architectures, and difficult for GBDT to exploit. Since the main cost of training of GBDT is gradient accumulation, it is hard to use low-precision FP resources such as FP16 in NVIDIA GPUs. This is because FP numbers have low accuracy for large values, simply accumulating gradients with FP16 would cause significant accumulation error. To achieve a meaningful speedup, we develop several novel techniques presented in the paper, including hierarchical histogram buffers, and packed gradient/hessian.
>
> Comparing to training with quantized gradients for other machine learning models like neural networks, this is well-motivated and highly non-trivial. Therefore, we believe that this is a significant novelty and contribution of our work.
>
> We appreciate the reviewer for spending time to review our paper and offer constructive suggestions. We hope that our response could address your concerns. If you are satisfied with our response, please kindly reconsider your score.
>
> [1] Seide F, Fu H, Droppo J, et al. 1-bit stochastic gradient descent and its application to data-parallel distributed training of speech dnns[C]//Fifteenth annual conference of the international speech communication association. 2014.
>
> [2] Wen W, Xu C, Yan F, et al. Terngrad: Ternary gradients to reduce communication in distributed deep learning[J]. Advances in neural information processing systems, 2017, 30.
>
> [3] Alistarh, Dan, et al. "QSGD: Communication-efficient SGD via gradient quantization and encoding." Advances in neural information processing systems 30 (2017).

---

### Official Review · Reviewer_vvMb · 2022-07-09

**Rating:** 4
**Confidence:** 2
**Soundness:** 2 fair
**Presentation:** 2 fair
**Contribution:** 2 fair

**Summary:**

This work proposes a low-precision training algorithm for GBDT based on gradient quantization and theoretical analyze the necessary precisions of gradients without hurting performance can be quite low. The authors also conduct extensive experiments on both CPUs and GPUs and the results show great improvements.



**Questions:**

1. Whether the quantized training of GBDT would have a negative influence on the inference performance ?

**Limitations:**

1. There are some other limitations that I think should be more discussed
2. The work is limited in its novelty.

**Strengths And Weaknesses:**

Pros:

1. The article is well written, quite clear.

Cons:
1. The advantages and limitations of the approach could be better underlined.
2. The work is limited in its novelty.

---

> ### Author Response · Authors · 2022-08-02
> **Response to Reviewer vvMb - Part 1**
>
> We sincerely thank the reviewer for reviewing our paper, and providing valuable suggestions. We address your concerns as follows:
>
> **Regrading the advantages.** The advantages of our work is mainly acclerating the training speed of GBDT without accuracy drop, which is an important topic and research direction to tree models. We will further stress the advanteages in the next version of our paper.
>
> **Regrading the limitations.** There is mainly one limitation of our approach: our theorem provides a strong guarantee under the assumption with enough data, however, when the number of data is small, empirically we found our method requires more bits to keep the accuracy. This limitation has been carefully discussed in section 7.1 (Line 282). We will further add more discussion to our paper about this limitation. Also, please kindly let us know if the reviewer think that there is any other limitation for our approach, and we will carefully discuss it in the future version of our paper.
>
> **Regrading the influence on inference performance.** Our theoretical analysis is for training loss. However, in the experiments, we report the accuracy on the test sets (We will make it clear in our next revision). All the accuracies reported in our paper are for test sets. As we can see from Table 2 of the paper, quantized training of GBDT preserves inference performance well.

---

> ### Author Response · Authors · 2022-08-02
> **Response to Reviewer vvMb - Part 2**
>
> We sincerely thank the reviewer for reviewing our paper, supporting our work, and providing valuable suggestions. We address your concerns as follows:
>
> **Regrading the novelty.** Training models with quantized gradients for neural networks are well-studied in previous literature but little is known on tree models. The differences appear in motivation, challenges, theoretical analysis, and implementations.
> - **Motivation.**
> (1). Training NN with quantized gradients aims to reduce the communication cost in distributed deep learning to speed up the training[1][2][3]. While both computation and communication cost could be largely reduced in training GBDT with quantized gradients. This is because that the computational bottleneck of NN are forward and backward propagation, which involves both weights and gradients, but the calculation of gradients in GBDT is quite fast, and the computation bottleneck is calculation of the statistics of gradients in histograms, which are also the main content of communication in distributed training. Therefore, gradient quantization would both accelerate training and communication in GBDT.
> (2). Quantized gradient slows the convergence in NN training because it enlarges the gradient's variance and will affect the convergence speed stochastic optimization algorithms. While it has little effect on training of GBDT since the model only care about the statistics of gradients, and the summation of stochastic quantization gradients are unbiased approximation to the summation of full-precision gradients. Therefore, training with gradient quantization is more well-motivated for GBDT models. We will fully discuss the differences in the next version of our paper.
>
> - **Challenges.** Training GBDT with quantized gradients faces additional challenges. For example, sometimes the stochastic quantization could substantially changes the splitting of the leaf, and brings intractable accuracy drop. To overcome such challenge, we identify stochastic quantization and leaf-value refitting are two important techniques. And the techniques are carefully verified both theoretically and empirically.
>
> - **Theoretical Analysis.** To best of our knowledge, we're the first to provide comprehensive theoretical analysis for training GBDT with quantized gradients. The analysis is novel and non-trivial. It is intuitive that with more data, the estimated summation with stochastic quantized gradient will be more accurate. However, to prove that the estimated split gain with quantized gradients is an asymptotically consistent estimator of the original split gain is difficult. We find that the key to achieve this is to consider the weak learnability assumption (Assumption 5.1).
>
> - **Implementations.** We provide a complete training framework for quantized GBDT with high efficiency in practical both on CPU and GPU architectures. It's quite challenging because that the support of low-precision calculation is limited for both architectures, and difficult for GBDT to exploit. Since the main cost of training of GBDT is gradient accumulation, it is hard to use low-precision FP resources such as FP16 in NVIDIA GPUs. This is because FP numbers have low accuracy for large values, simply accumulating gradients with FP16 would cause significant accumulation error. To achieve a meaningful speedup, we develop several novel techniques presented in the paper, including hierarchical histogram buffers, and packed gradient/hessian.
>
> Comparing to training with quantized gradients for other machine learning models like neural networks, this is well-motivated and highly non-trivial. Therefore, we believe that this is a significant novelty and contribution of our work.
>
> We appreciate the reviewer for spending time to review our paper and offer constructive suggestions. We hope that our response could address your concerns. If you are satisfied with our response, please kindly reconsider your score.
>
> [1] Seide F, Fu H, Droppo J, et al. 1-bit stochastic gradient descent and its application to data-parallel distributed training of speech dnns[C]//Fifteenth annual conference of the international speech communication association. 2014.
>
> [2] Wen W, Xu C, Yan F, et al. Terngrad: Ternary gradients to reduce communication in distributed deep learning[J]. Advances in neural information processing systems, 2017, 30.
>
> [3] Alistarh, Dan, et al. "QSGD: Communication-efficient SGD via gradient quantization and encoding." Advances in neural information processing systems 30 (2017).

---

### Official Review · Reviewer_PPQi · 2022-07-10

**Rating:** 7
**Confidence:** 4
**Soundness:** 3 good
**Presentation:** 4 excellent
**Contribution:** 3 good

**Summary:**

This papers proposes a way to speedup the training of GBDTs via quantization technique. They adopt the idea of quantized gradients that is widely used in neural net training/compression literature and show that it is possible to reduce the precision as low as 2-3 bits and yet to achieve similar performance in terms of accuracy. The overall speedup in training time can be up to 2x.

**Questions:**

.

**Limitations:**

.

**Strengths And Weaknesses:**

Strengths:
- Practical and simple approach to speedup GBDT training for both CPU and GPU;
- The paper is clearly written with nice introduction to GBDT training preliminaries and techniques which is especially useful for non-experts
- Section 6 describes system implementation details which is quite interesting. I especially liked the idea of packing the gradients and hessian.
- It is quite surprising to see that lowering the precision that much (up to 2-3 bit) would preserve the accuracy. It definitely has research/practical value and raises an interesting question: are GBDTs overparameterized? It is not a secret that the ensemble size in production environment can be huge. And if that is the case, then should we consider compressing them?

Weaknesses:
- Authors provide theoretical analysis of the method (i.e., error bound after quantization) which is great. However, there are some limitations: 1) the main theorem is only valid decision stumps; 2) the bound is presented for EACH GRADIENT. How about their summations or a bound for histogram counts? Moreover, the error can be proportional to max|$g_i$| (plus square of it) which I believe is pretty high and not sure if it is useful result.
- Novelty. The training of GBDTs is done in traditional way: gradient+hessian, feature histograms, etc. So, there is no contribution from that angle. The application of quantization here is also straightforward and done via binning and rounding. However, it seems there is no previous work which combines them, whereas this paper shows its huge benefits.


----- After Rebuttal ------------

My main concerns regarding this paper were addressed during rebuttal and I'm willing to increase my score to 7. Overall, I think that this paper is a nice contribution with both theoretical and practical results. Especially, experimental results suggest strong implications: one can significantly quantize (compress) gradients and still learn pretty accurate boosted trees. Novelty seems to be fine: although quantization is well known technique in NN literature, this paper seems to be the first who proposes applying it to trees and did a nice job on that.

---

> ### Author Response · Authors · 2022-08-02
> **Response to Reviewer PPQi - Part 1**
>
> We sincerely thank the reviewer for reviewing our paper, supporting our work, and providing valuable suggestions. We address your concerns as follows:
>
> **Regrading the main theorem.** In Theorem 5.3, our analysis is not specialized for decision stumps. The theorem considers a single split within a tree of any size. Our definition of weak learnability is for stumps, which may cause confusion (we will clarify this in our revision). But in our theorem, the assumption is applied to the stump with node $s$ as root, where node $s$ can be any node within a decision tree of arbitrary size. Thus Theorem 5.3 states that every single split during the tree growing process is well approximated with quantized gradients, regardless of the tree size.
>
> **Regrading the bound.** Theorem 5.3 is not presenting the bound for each gradient. The curlycue $\mathcal{G}$ in the theorem represents the split gain, plus the optimal loss in leaf $s$ (which is constant w.r.t. different splits of leaf $s$), as is defined in line 207 of the paper. Thus $\mathcal{G}$ is the criterion for choosing the split in leaf $s$ for decision tree training. And according to equation (3) in the paper, $\mathcal{G}$ is calculated with the summation of gradients and counts from histograms. Thus Theorem 5.3 depicts the effects of quantization on the value of split selection criteria for leaf $s$.
>
> **Regrading the scale of $\max |g_i|$.** For the scale of $\max{\left\vert g_i \right\vert}$, as we stated in the paper (line $220\sim221$), it is bounded for classification problems. For regression, the magnitude of gradients, which is influenced by the magnitude of labels, is dataset specific. But we can always scale the original labels to a fixed range, e.g., $[-1, 1]$.
>
> **Regrading the upper bound.** To further address your concern about the upper bound, here we calculate the upper bound in the second conclusion of Theorem 5.3 with an example case. Suppose we are handling a regression task. For square loss, $\max \left\vert g_i \right\vert$ is at most the maximum magnitude of the label. Let $\epsilon = \frac{\max \left\vert g_i \right\vert}{10}$, and with a learning rate $0.1$ (which is the setting in our experiment). Then if the first conclusion in Theorem 5.3 holds, the average change of prediction values for data in leaf $s$ and its descendants after one epoch would be no more than $\frac{\max{\left\vert g_i \right\vert}}{100}$. If the second conclusion in Theorem 5.3 holds, let $\delta = 0.1$, and suppose that in leaf $s$ we have $\gamma_s=0.2$, then the upper bound would be approximately
> $$
> \frac{679}{2^{B-1}}\left( \sqrt{\frac{1}{n_{s_1}}} + \sqrt{\frac{1}{n_{s_2}}} \right) + \frac{9222}{n_s \cdot 4^{B-2}},
> $$
> Suppose that we have $10^7$ data points in leaf $s$, and the split is not too imbalanced such that $s_1$ and $s_2$ each have at least $10^6$ data points, then the following table shows values of the upper bound with different gradient bits $B$. If we use $4$-bit gradients, then with a probability of at least $90$ percents, the error of split gain for any split in leaf $s$, relative to the largest split gain of leaf $s$, is no more than $17\%$. And with $8$-bit gradients, the value is about $1\%$.
> |  $B$   | 2  | 3  | 4  | 5  | 8  |
> |  ----  | ----  | ----  | ----  | ----  | ----  |
> | bound value  | 0.679922 |0.339731 |0.169808 |0.084889 |0.010610 |
>
> As we can see the error caused by quantization in split gain estimation decreases exponentially with the number of gradient bits.

---

> ### Author Response · Authors · 2022-08-02
> **Response to Reviewer PPQi - Part 2**
>
> We sincerely thank the reviewer for reviewing our paper, supporting our work, and providing valuable suggestions. We address your concerns as follows:
>
> **Regrading the novelty.** Training models with quantized gradients for neural networks are well-studied in previous literature but little is known on tree models. The differences appear in motivation, challenges, theoretical analysis, and implementations.
> - **Motivation.**
> (1). Training NN with quantized gradients aims to reduce the communication cost in distributed deep learning to speed up the training[1][2][3]. While both computation and communication cost could be largely reduced in training GBDT with quantized gradients. This is because that the computational bottleneck of NN are forward and backward propagation, which involves both weights and gradients, but the calculation of gradients in GBDT is quite fast, and the computation bottleneck is calculation of the statistics of gradients in histograms, which are also the main content of communication in distributed training. Therefore, gradient quantization would both accelerate training and communication in GBDT.
> (2). Quantized gradient slows the convergence in NN training because it enlarges the gradient's variance and will affect the convergence speed stochastic optimization algorithms. While it has little effect on training of GBDT since the model only care about the statistics of gradients, and the summation of stochastic quantization gradients are unbiased approximation to the summation of full-precision gradients. Therefore, training with gradient quantization is more well-motivated for GBDT models. We will fully discuss the differences in the next version of our paper.
>
> - **Challenges.** Training GBDT with quantized gradients faces additional challenges. For example, sometimes the stochastic quantization could substantially changes the splitting of the leaf, and brings intractable accuracy drop. To overcome such challenge, we identify stochastic quantization and leaf-value refitting are two important techniques. And the techniques are carefully verified both theoretically and empirically.
>
> - **Theoretical Analysis.** To best of our knowledge, we're the first to provide comprehensive theoretical analysis for training GBDT with quantized gradients. The analysis is novel and non-trivial. It is intuitive that with more data, the estimated summation with stochastic quantized gradient will be more accurate. However, to prove that the estimated split gain with quantized gradients is an asymptotically consistent estimator of the original split gain is difficult. We find that the key to achieve this is to consider the weak learnability assumption (Assumption 5.1).
>
> - **Implementations.** We provide a complete training framework for quantized GBDT with high efficiency in practical both on CPU and GPU architectures. It's quite challenging because that the support of low-precision calculation is limited for both architectures, and difficult for GBDT to exploit. Since the main cost of training of GBDT is gradient accumulation, it is hard to use low-precision FP resources such as FP16 in NVIDIA GPUs. This is because FP numbers have low accuracy for large values, simply accumulating gradients with FP16 would cause significant accumulation error. To achieve a meaningful speedup, we develop several novel techniques presented in the paper, including hierarchical histogram buffers, and packed gradient/hessian.
>
> Comparing to training with quantized gradients for other machine learning models like neural networks, this is well-motivated and highly non-trivial. Therefore, we believe that this is a significant novelty and contribution of our work.
>
> We appreciate the reviewer for spending time to review our paper and offer constructive suggestions. We hope that our response could address your concerns. If you are satisfied with our response, please kindly reconsider your score.
>
> [1] Seide F, Fu H, Droppo J, et al. 1-bit stochastic gradient descent and its application to data-parallel distributed training of speech dnns[C]//Fifteenth annual conference of the international speech communication association. 2014.
>
> [2] Wen W, Xu C, Yan F, et al. Terngrad: Ternary gradients to reduce communication in distributed deep learning[J]. Advances in neural information processing systems, 2017, 30.
>
> [3] Alistarh, Dan, et al. "QSGD: Communication-efficient SGD via gradient quantization and encoding." Advances in neural information processing systems 30 (2017).

---

### Meta-Review · Area_Chair_9ZpX · 2022-08-31

**Recommendation:** Accept
**Confidence:** Less certain

**Metareview:**

The reviewers conclude on an interesting paper with broad messaging that does make sense -- I do subscribe to it as well. I recommend it for acceptance, noting that interactions with reviewers were the occasion to provide additional remarks that have to be used to craft the camera ready, in particular for the remarks made to PPQi (mostly the technical part of this discussion).

**Award:**

No

---

### Decision · Program_Chairs · 2022-09-14

Accept